chemical engineering

calcium carbide, acetylene hydrogenation, ethylene, Pd/MCM-41 catalyst

**Author for correspondence:**
Mingyuan Zhu
e-mail: zhuminyuan@shzu.edu.cn

# Pd/MCM-41 catalyst for acetylene hydrogenation to ethylene

## Lihua Kang[1], Bozhen Cheng[2] and Mingyuan Zhu[1,2]

[1]College of Chemistry and Chemical Engineering, Yantai University, Yantai, Shandong 264005, People's Republic of China
[2]School of Chemistry and Chemical Engineering, Shihezi University, Shihezi, Xinjiang 832003, People's Republic of China

(iD) LK, 0000-0002-0059-7877; MZ, 0000-0003-1181-6358

This study aims to produce ethylene using the calcium carbide route, acetylene from calcium carbide and then selective hydrogenation of the high-concentration acetylene to ethylene. A series of catalysts with different supports, such as $Al_2O_3$, $SiO_2$ and MCM-41, were prepared using the ethylene glycol reduction method and their catalytic properties for high-concentration acetylene hydrogenation of calcium carbide to ethylene were studied by transmission electron microscopy, X-ray powder diffraction and thermogravimetry, among others. The results show that the small particle size and uniform dispersion of palladium (Pd) particles in the Pd/MCM-41 catalyst produced the highest ethylene yield of 62.09%. Then, the conditions for the basic reaction, such as reaction temperature and space velocity, were optimized using MCM-41 as a support. The yield of ethylene after condition optimization was as high as 82.87%, while the loading of Pd was 0.1%.

## 1. Introduction

Ethylene is an important chemical product and the demand for ethylene has increased annually. The traditional method of producing ethylene is hydrocarbons cracking [1–3]. However, this production process is greatly limited by the cost and the nature of crude oil. Calcium carbide was formed by the reaction of quicklime and coal at around 2000°C [4], and acetylene was produced by reacting calcium carbide and water. Calcium carbide could be directly used as an easy-to-handle and efficient source of acetylene, and acetylene gas was safe and non-explosive [5,6]. Due to the low cost of coal and quicklime raw materials, acetylene from the calcium carbide route was easily available and inexpensive [7], especially in regions that have plenty of coal such as China. The production of ethylene from acetylene hydrogenation may be a good choice due to the significance of ethylene and abundance of acetylene. In this

route, acetylene is prepared by calcium carbide, and then further hydrogenation for the high-concentration acetylene to ethylene. This route is of great significance to expand the downstream industrial chain of acetylene and to make full use of coal [8].

However, the investigation of pure acetylene hydrogenation was scarce, and most investigation focused on the removal process of trace acetylene in the production of ethylene (trace acetylene hydrogenation to ethylene) [9,10]. Supported palladium (Pd)-based catalysts have been widely used and studied in the field of trace acetylene hydrogenation owing to their excellent hydrogenation catalytic activity [11,12]. Typically, the catalytic activity of Pd-based catalysts is further enhanced by the size adjustment, ligand modification and second metal modification [13,14], facilitating the reaction towards path I. In addition, the support for a Pd catalyst also has an important influence on the activity of the catalyst [15,16]. First, the nature of support can affect the dispersibility of the metal particles. He $et$ $al$. prepared Pd-based catalysts using hydrotalcite (HT), MgO and $Al_2O_3$ as supports, respectively. It was found that Pd atoms re-dispersed under the influence of acidic sites on support, forming a large number of active sites and resulting in a significantly higher Pd catalyst activity than other catalysts [17]. Panpranot $et$ $al$. [18] also found that Pd/MCM-41 macroporous catalysts have higher Pd dispersion and higher hydrogenation activity and lower metal loss than $Pd/SiO_2$ catalysts. Also, the interaction between the active components and support can also affect the catalytic activity for acetylene hydrogenation using Pd catalysts. Panpranot $et$ $al$. prepared $Pd/TiO_2$ catalysts having different crystal phases of $TiO_2$ as supports. It was found that the contact of $Ti^{3+}$ with Pd reduced the adsorption strength of ethylene and improved the selectivity of ethylene; consequently, the $TiO_2$ containing 44% rutile was the optimal support for the Pd catalyst [19]. Researchers also prepared PdAg/NiTi-LDH catalyst that showed superior hydrogenation activity than other catalysts because the presence of $Ti^{3+}$ defect sites and the high dispersion of metal particles enhanced the dissociation and activation of $H_2$ [20].

It must be noted that the catalytic performance of acetylene hydrogenation via the calcium carbide route may be different from these catalysts because almost pure acetylene is used in the catalytic reaction process. It is quite different from the hydrogenation of a trace amount of acetylene in ethylene. Here, we tried to explore an active catalyst and applied it to the acetylene hydrogenation via the calcium carbide route. MCM-41 was selected as the support for the Pd catalyst and compared with $SiO_2$ and $Al_2O_3$ due to its unique skeletal structure elements and pore size control. The effects of the acidity and specific surface area of the support on the catalytic performance were also investigated. The reaction condition of the Pd catalyst was optimized to provide basic data for the industrial process.

# 2. Experimental section

## 2.1. Catalyst preparation

A series of Pd catalysts with different supports, such as $SiO_2$, $Al_2O_3$ and MCM-41, were prepared by ethylene glycol reduction. First, 21.1 mg of $Pd(OAc)_2$ was dissolved in hydrochloric acid solution and mixed with 50 ml of ethylene glycol solution, before the support was added by vigorous stirring. Second, approximately 2.2 mmol of NaOH solution was added dropwise to adjust the pH value to 11. The solution was then heated to 130°C for 6 h and was cooled to room temperature. Finally, the suspension was filtered, washed and dried under vacuum at 100°C for 24 h. The theoretical loading of Pd was calculated as 1 wt%.

## 2.2. Catalyst characterization

The specific surface area and the pore structure (Brunauer–Emmett–Teller; BET) were analysed by Micromeritics ASAP 2020C. X-ray diffraction (XRD) data were collected by a Bruker D8 Advance X-ray diffractometer with Cu Kα radiation at an acceleration voltage of 40 kV and a current density of 40 mA. The transmission electron microscopic (TEM) images were analysed by Tecnai G2 F20 TEM (200 kV). ICAP6300 was performed to gather the inductively coupled plasma–atomic emission spectrometry (ICP-AES) results and confirm the actual metal loadings of the catalysts. NETZSCH STA 449F3 Jupiter was chosen to collect thermogravimetric analysis (TG) and derivative thermogravimetry (DTG) data and analyse carbon deposition on the catalyst. AUTOChem II 2920 was performed to collect $NH_3$-TPD data and analyse the properties of supports. Before the test, the sample was purged under an inert atmosphere. X-ray photoelectron spectroscopy (XPS) analyses were performed using an ESCA 3400 (Kratos Analytical Ltd, Manchester, UK).

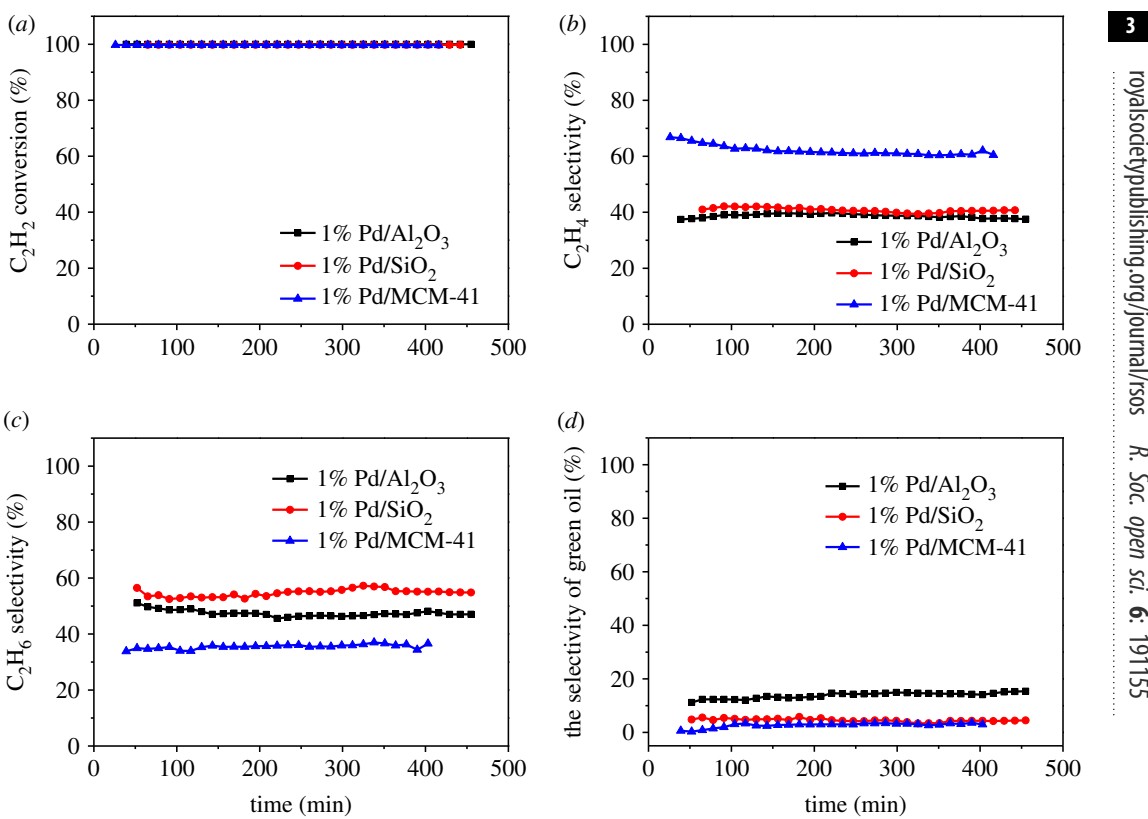

**Figure 1.** (*a*) Acetylene conversion, (*b*) ethylene selectivity, (*c*) ethane selectivity and (*d*) green oil selectivity of Pd catalysts with different supports ($GHSV(C_2H_2) = 2318 \text{ h}^{-1}$, $T = 150°C$ and $V(H_2) : V(C_2H_2) = 2 : 1$).

## 2.3. Catalytic performance evaluation

The reaction was run on a small fixed-bed catalytic reactor with a stainless steel reaction tube (i.d. = 10 mm). The feed gas was only acetylene (99.99%) and hydrogen (99.999%) without dilution gas, both the acetylene and hydrogen gases were obtained from the cylinder directly. Prior to the experiment, the catalyst was pre-treated at 130°C for 2 h under $H_2$ atmosphere at a flow rate of 80 ml min$^{-1}$ to remove the moisture on the catalyst. The export gas was analysed using online gas chromatography (Shimadzu, GC-2014) equipped with a TCD detector and Porapak-N column (2.1 mm × 2 m). The chromatogram results showed four major peaks, which have been experimentally verified to be those of hydrogen, ethylene, ethane and acetylene, without other distinct peaks. It was indicated that the gaseous product mainly comprised $H_2$, $C_2H_4$, $C_2H_6$ and $C_2H_2$. Thus, the conversion of $C_2H_2$ and the selectivity of $C_2H_4$, $C_2H_6$ and green oil were calculated using the following formulae:

$$C_2H_2 \text{ conversion} = \frac{C_2H_2(\text{feed}) - C_2H_2(\text{out})}{C_2H_2(\text{feed})} \times 100\%, \tag{2.1}$$

$$C_2H_4 \text{ selectivity} = \frac{C_2H_4(\text{out})}{C_2H_2(\text{feed}) - C_2H_2(\text{out})} \times 100\%, \tag{2.2}$$

$$C_2H_6 \text{ selectivity} = \frac{C_2H_6(\text{out})}{C_2H_2(\text{feed}) - C_2H_2(\text{out})} \times 100\% \tag{2.3}$$

and

$$\text{the selectivity of green oil} = \left(1 - \frac{C_2H_4(\text{out}) + C_2H_6(\text{out})}{C_2H_2(\text{feed}) - C_2H_2(\text{out})}\right) \times 100\%. \tag{2.4}$$

# 3. Results and discussion

## 3.1. Effect of different supports on catalytic performance

The performance of Pd catalysts with $SiO_2$, $Al_2O_3$ and MCM-41 as supports is shown in figure 1. According to figure 1*a*, the $C_2H_2$ conversion of all catalysts almost reached 100%. The $C_2H_4$ selectivity

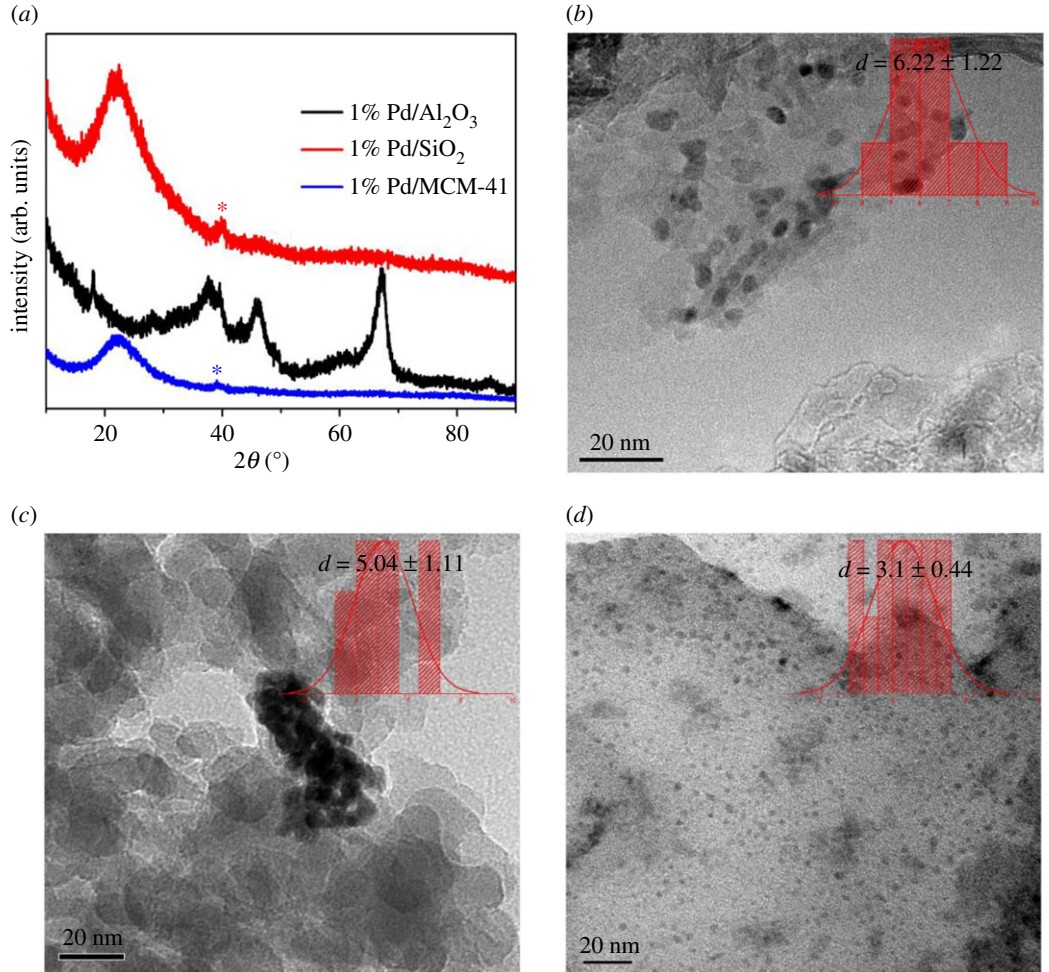

**Figure 2.** (a) XRD patterns of Pd catalysts with different supports and TEM images of (b) 1% Pd/Al₂O₃, (c) 1% Pd/SiO₂ and (d) 1% Pd/MCM-41.

of the 1% Pd/MCM-41 catalyst was the highest (62.09%), following by approximately 40% $C_2H_4$ selectivity of 1% Pd/SiO₂ and Pd/Al₂O₃ catalysts (figure 1$b$). Meanwhile, 1% Pd/MCM-41 catalyst had the lowest $C_2H_6$ selectivity of approximately 35% (figure 1$c$). It was only 47% for the 1% Pd/Al₂O₃ catalyst and was highest at 55% for the Pd/SiO₂ catalyst. In the process of acetylene hydrogenation via the calcium carbide route, the study on green oil is particularly necessary because the high-concentration acetylene is easily polymerized under unfavourable conditions to form green oil covering the surface of the catalyst, resulting in deactivation of the catalyst [21,22]. Therefore, we calculated the selectivity of green oil. As shown in figure 1$d$, the selectivity of green oil was the highest on the 1% Pd/Al₂O₃ catalyst, reaching 14%, which was only approximately 4% and 3% on Pd/SiO₂ and Pd/MCM-41 catalysts, respectively.

To illustrate the performance difference of the synthesized 1% Pd/SiO₂, Pd/Al₂O₃ and Pd/MCM-41 catalysts, we performed a series of characterization of this Pd-based catalyst. XRD patterns of different catalysts are shown in figure 2$a$. It is clear to see the diffraction peaks of Pd atoms on three catalysts at approximately 39°, 45°, 67° and 81°, corresponding to Pd (111), (200), (220) and (311) crystal face, respectively, which is consistent with the typical crystalline Pd face-centred cubic [23]. It must be noted that the peak of Pd on the Pd/Al₂O₃ catalyst was unclear, possibly because the diffraction peaks of Al₂O₃ and Pd atoms were close. Meanwhile, we found that the diffraction peaks of Pd/SiO₂ catalysts were sharper than those on the Pd/MCM-41 catalyst. This phenomenon may reveal that the particle size on MCM-41 was smaller than the other supports.

To testify this hypothesis, we further characterized the distribution and particle size of Pd. It can be seen from the TEM images in figure 2$b$–$d$ that the Pd nanoparticles on the Pd/MCM-41 catalyst had the smallest particle size of approximately 3.10 nm and the best uniform dispersion among these three catalysts. The average particle size of the 1% Pd/Al₂O₃ catalyst was approximately 6.22 nm with

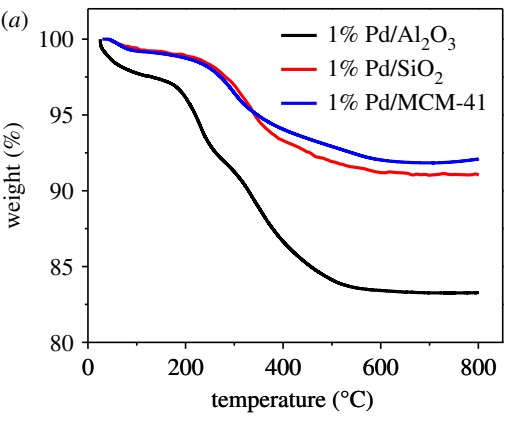 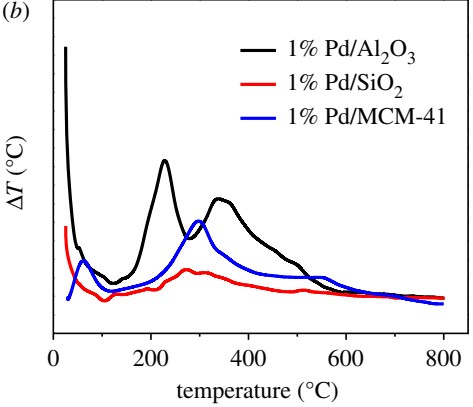

**Figure 3.** (*a*) TG and (*b*) DTG results of 1% Pd/Al$_2$O$_3$, Pd/SiO$_2$ and Pd/MCM-41 after 7.5 h reaction.

**Table 1.** N$_2$ sorption characteristic of different supports.

| samples | surface area (m$^2$ g$^{-1}$) | pore volume (cm$^3$ g$^{-1}$) | average pore size (nm) |
|---|---|---|---|
| Al$_2$O$_3$ | 203.2 | 0.4 | 6.1 |
| SiO$_2$ | 203.8 | 0.5 | 12.6 |
| MCM-41 | 1163.8 | 1.2 | 3.2 |

slight agglomeration. The average particle size of the 1% Pd/SiO$_2$ catalyst was 5.04 nm with significant agglomeration. The high dispersion of metal Pd could inhibit the formation of the β-PdH phase because it promotes more ethane formation [24,25]. Thus, a large particle size and agglomeration of metal particles in Pd/Al$_2$O$_3$ and Pd/SiO$_2$ catalysts make them less selective for ethylene. The smaller particle size of approximately 3.10 nm and better uniform dispersion of Pd on the surface of MCM-41 may be one of the reasons for its better C$_2$H$_4$ selectivity.

It was well known that the dispersion of metal particles is related to the specific surface area of a support. We characterized the surface area, pore volume and pore size of Al$_2$O$_3$, SiO$_2$ and MCM-41 supports, respectively. As listed in table 1, Al$_2$O$_3$ has the smaller specific surface area and the medium pore structure. SiO$_2$ has the smaller specific surface area and the largest pore size. MCM-41 owns the largest BET surface area and pore volume, smallest pore size. According to the research [26,27], as the pore size increases, the dispersion of metal particles on the surface of support becomes smaller, and the particle is larger. Lonergan *et al.*'s [28] research also reported that metal nanoparticles on a high specific surface area of γ-Al$_2$O$_3$ and ZrO$_2$ were lower than those on a low surface area of α-Al$_2$O$_3$ and ZrO$_2$. Therefore, we conclude that the poor BET surface area and large pore size on Al$_2$O$_3$ and SiO$_2$ result in the metal particles agglomeration and large particle size, while the high BET surface area and the ordered pore structure of MCM-41 make Pd particles the most uniformly distributed on MCM-41 with the smallest particle size, among the three catalysts.

Figure 1 shows the selectivity of green oil was somewhat considerable. Therefore, we studied the amount of carbonaceous deposits after the reaction of different catalysts for 7.5 h by TG. As shown in figure 3*a*, in the temperature range of 0–100°C, the mass loss of the catalyst occurred due to the evaporation of water on the surface, whereas in the range of 100–500°C, the decomposition of the polymer led to a reduction in the mass. The mass loss of the Pd/Al$_2$O$_3$ catalyst was up to 17%, and those of Pd/MCM-41 and Pd/SiO$_2$ catalysts were approximately 6% and 8%, respectively. It is obvious that the 1% Pd/Al$_2$O$_3$ catalyst had the most carbon deposition. As shown in figure 3*b*, only a significant peak occurred at around 300°C for Pd/MCM-41 and Pd/SiO$_2$ catalysts due to carbon deposition on or near Pd particles and partial combustion of heavy hydrocarbons adsorbed on the catalyst. However, for the Pd/Al$_2$O$_3$ catalyst, there were two distinct peaks in the DTG curve. The first peak was at 220°C mainly due to heavy hydrocarbons adsorbed on the catalyst surface or pores. The second peak was around 350°C mainly due to the combustion of carbon on the Pd particles or near the Pd [29]. According to previous literature [30], the carbon deposit near 370°C was a precursor of graphitic carbon, which covers the surface of the support and its accumulation can cause considerable

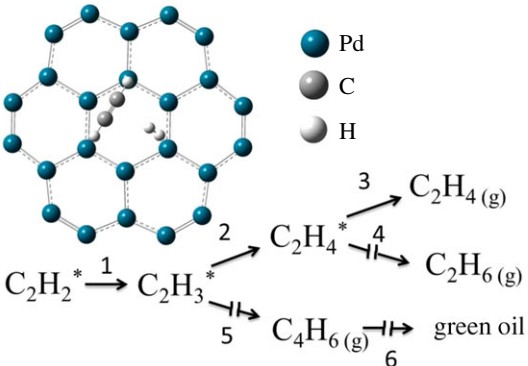

**Figure 4** Reaction mechanism of acetylene hydrogenation.

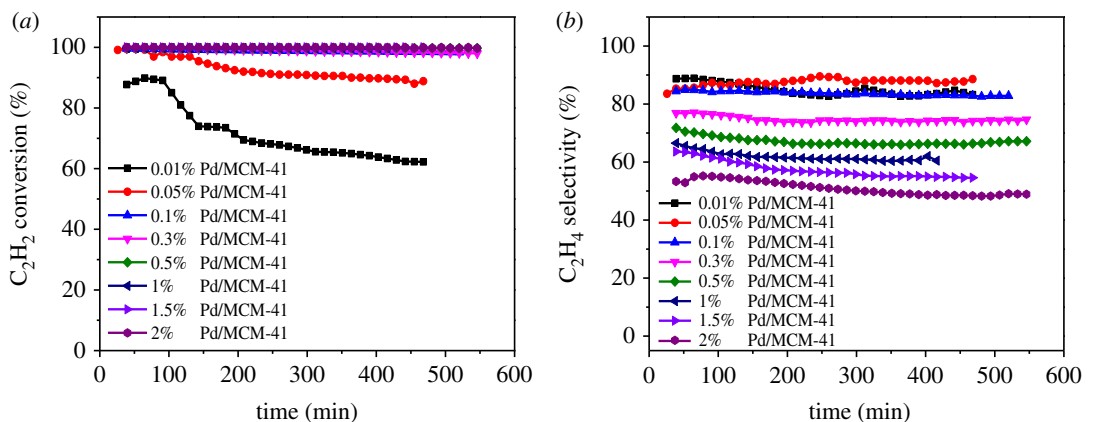

**Figure 5.** Optimization of Pd wt% (*a*) acetylene conversion and (*b*) ethylene selectivity.

space barriers. The steric hindrance caused by the precursor of graphitic carbon, and the heavy hydrocarbons adsorbed on the surface or pores of the catalyst lead to excessive hydrogenation, resulting in low $C_2H_4$ selectivity of the 1% $Pd/Al_2O_3$ catalyst.

Figure 4 gives the plausible reaction mechanism of acetylene hydrogenation on the surface of the Pd catalyst. As shown in figure 4, the reactant acetylene and hydrogen adsorbed on the surface of active Pd sites and hydrogen dissociated into H atom. Adsorbed $C_2H_2$* reacted with one H atom to form $C_2H_3$* intermediates (Step 1). If no H atom was provided, $C_2H_3$* would react with its dimer to produce green oil by side reactions (Steps 5 and 6). This side reaction was an undesired reaction because excessive acetylene in the reaction tends to polymerize and the resulting polymer covers the active sites, thus reducing the stability of the catalyst. If there were enough H atom to react with $C_2H_3$* intermediates to produce $C_2H_4$* (Step 2), then $C_2H_4$ gas could be obtained by a desorption process (Step 3). This was the desired reaction which produces more ethylene. When under adverse reaction conditions, excessive hydrogenation of adsorbed ethylene leads to the direct hydrogenation of acetylene to ethane by strong adsorption species (Step 4).

## 3.2. Optimization of reaction conditions

The Pd/MCM-41 catalyst showed better catalytic performance than $Pd/Al_2O_3$ and $Pd/SiO_2$ catalysts. We optimized the loading of Pd on MCM-41 support in the next investigation. Figure 5*a,b* shows the optimization of the loading of Pd (wt%) at a condition of $V(H_2):V(C_2H_2)=2:1$, gas hourly space velocity (GHSV) $(C_2H_2)=2318\ h^{-1}$ and $T=150°C$. It is obvious that when the loading of Pd was 0.1%, the yield of $C_2H_4$ was up to 82.74%. It can be clearly observed that the $C_2H_2$ conversion reduced by a decrease in the Pd content, whereas the selectivity of $C_2H_4$ showed an opposite trend. This phenomenon is consistent with the literature [31].

Figure 6*a,b* shows the optimization of the GHSV($C_2H_2$) at a condition of $T=150°C$, $V(H_2):V(C_2H_2)=2:1$ on the 0.1% Pd/MCM-41 catalyst. When GHSV was 2318 $h^{-1}$, the yield of $C_2H_4$ reached 82.74%. Obviously, the acetylene conversion decreased as the GHSV increases. When the GHSV was too high, the residence

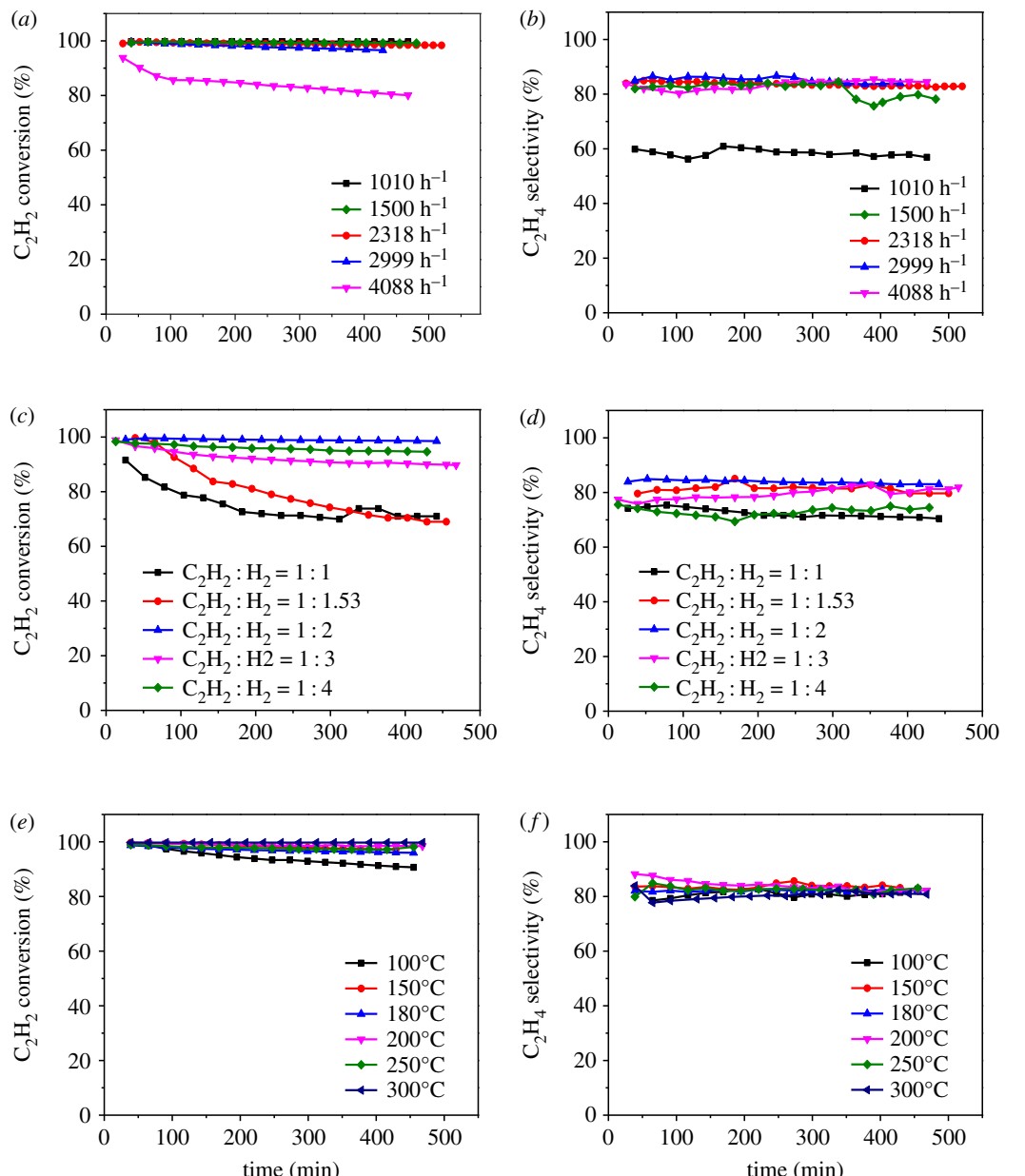

**Figure 6.** (a,c,e) Acetylene conversion and (b,d,f) ethylene selectivity; optimization of (a,b) GHSV, (c,d) V(H$_2$):V(C$_2$H$_2$) and (e,f) reaction temperature.

time of reactants was reduced, resulting in a significant decrease in the conversion of acetylene [32,33]. Figure 6c,d presents the optimization of the volume ratio of C$_2$H$_2$ and H$_2$. We found that by increasing the volume ratio, the C$_2$H$_2$ conversion decreased, which is consistent with the literature [34]. Figure 6e,f shows the screening of the reaction temperature. Although the C$_2$H$_4$ selectivity was very close at different temperatures, there was a significant temperature rise during the reaction because the acetylene hydrogenation reaction is exothermic. Therefore, we chose 200°C as the reaction temperature, and when the reaction temperature reached 200°C, the C$_2$H$_4$ yield remained at 82.87%. It can be clearly observed that the acetylene conversion was only 90% at 100°C, indicating that a low temperature is not conducive to the conversion. This result corresponds to that in previous literature [32,34].

## 3.3. Stability test

Based on these experimental results, we found that the 0.1% Pd/MCM-41 catalyst has a significant catalytic activity at the optimization conditions. Therefore, we tested the stability of the catalyst under these conditions. As shown in figure 7, after 32 h of reaction, the conversion of the catalyst began to

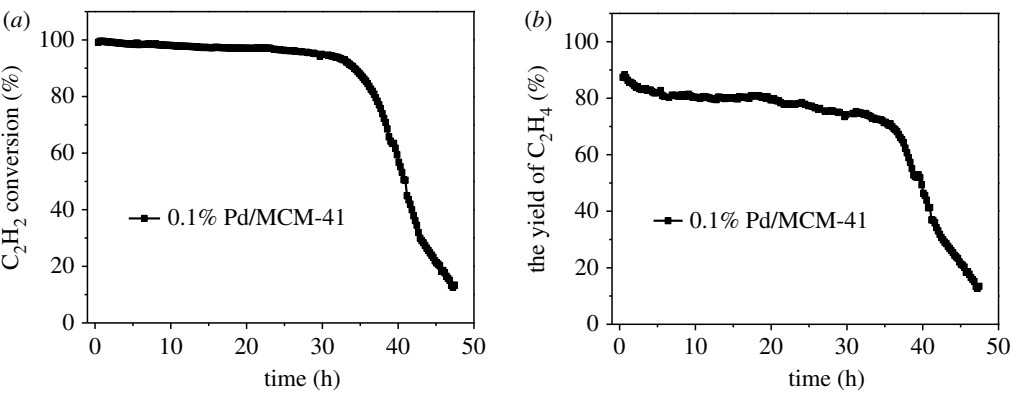

**Figure 7.** Stability test of 0.1% Pd/MCM-41 (*a*) acetylene conversion and (*b*) ethylene yield.

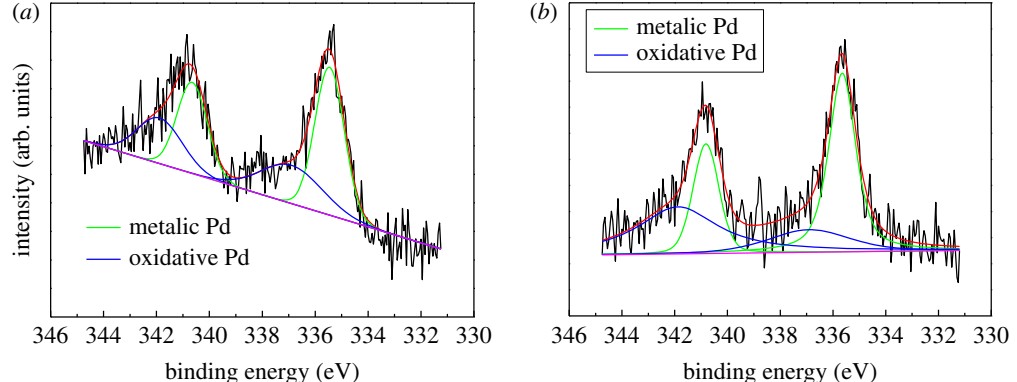

**Figure 8.** The Pd 3d regions of the XPS spectra of fresh 0.1% Pd/MCM-41 and spent 0.1% Pd/MCM-41.

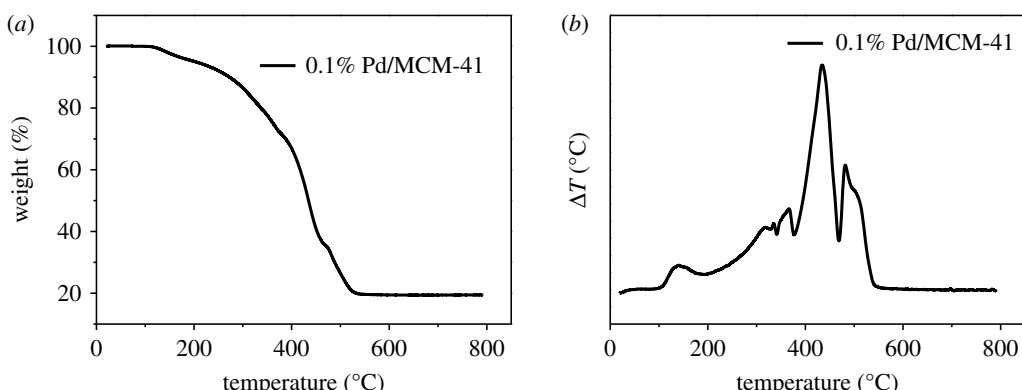

**Figure 9.** (*a*) TG and (*b*) DTG results of 0.1% Pd/MCM-41 after stability test.

drop sharply. After the reaction was carried out for 47 h, the catalyst was substantially deactivated with only 13% $C_2H_2$ conversion and about 12% $C_2H_4$ yield.

In order to investigate the reason for deactivation of the 0.1% Pd/MCM-41 catalyst, ICP-AES was carried out to determine the loading of Pd before and after the stability test. The actual loading of Pd was 0.096% and 0.094% in the fresh and spent Pd/MCM-41 catalyst, respectively. Not much difference of Pd loading was found in the fresh and spent Pd/MCM-41 catalyst, indicating that no metal leaching occurred during the hydrogen process of acetylene. XPS experiments were carried out to identify the change of the oxidation state of Pd in the 0.1% Pd/MCM-41 catalyst before and after the stability test, and the results are shown in figure 8. Calculated from the XPS peaks of Pd(0) and Pd(II), the ratio of oxidative Pd(II) in the fresh 0.1% Pd/MCM-41 was about 35.2%, and it reduced to 22.7% after 32 h stability test, which indicated that the part of oxidative Pd(II) was reduced to metallic Pd(0) under the reaction atmosphere.

Figure 9*a* shows the mass loss of the Pd/MCM-41 catalyst after stability (about 80.4%), with mass loss mainly due to polymer consumption. According to figure 9*b*, there are mainly three peaks at

approximately 360, 435 and 480°C. Combined with the result of figure 3, the catalyst was gradually deactivated due to carbon deposition near the Pd or Pd covering a portion of the active sites.

# 4. Conclusion

Pd/MCM-41 exhibited the excellent catalytic performance for acetylene hydrogenation via the calcium carbide route. The optimal metal loading was 0.1%, and the catalyst showed the best catalytic performance under the reaction condition of GHSV = 2318 h$^{-1}$, V($C_2H_2$) : V($H_2$) = 1 : 2 and $T$ = 200°C. It was proposed that the excellent catalytic performance of Pd/MCM-41 was ascribed to the good dispersion and small particle size of Pd due to the high surface area and weak acidity of MCM-41 support. Pd/MCM-41 catalyst might be a candidate catalyst for the acetylene hydrogenation via the calcium carbide route.

Data accessibility. All original data are deposited at the Dryad Digital Repository: http://dx.doi.org/10.5061/dryad. vs37881 [35].

Authors' contributions. M.Z. designed and conceived the experiments; L.K. performed the experiments, analysed the data and wrote the paper. B.C. carried out the characterization of the catalysts. All authors gave final approval for publication.

Competing interests. We have no competing interests.

Funding. This work was supported by the State Key Research and Development Project of China (grant no. 2016YFB0301603), the International Corporation of S&T Project in Xinjiang Bingtuan (grant no. 2018BC003) and the International Corporation of S&T Project in Shihezi University (grant no. GJHZ201701).

Acknowledgements. We thank Prof. Bin Dai of Shihezi University for his guidance in the experimental progress.

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
