## [Reviewer comments · Royal Society Open Science]

Review History

RSOS-191155.R0 (Original submission)

Review form: Reviewer 1 (Ummadisetti Chinnarajesh)

Is the manuscript scientifically sound in its present form?

No

Are the interpretations and conclusions justified by the results?

No

Is the language acceptable?

No

Do you have any ethical concerns with this paper?

No

Have you any concerns about statistical analyses in this paper?

No

Recommendation?

Reject

Comments to the Author(s)

Zhu et al. reported Pd/MCM-41 catalyst for acetylene hydrogenation to ethylene via calcium carbide route. However, the hydrogenation of acetylene gas to ethylene using various metal oxide supported Pd satellite sites are well documented in the literature. The merit of present work is the utilization of in-situ generated acetylene feedstock from an inexpensive and renewable calcium carbide. Although, authors have neither discussed the context in the introduction nor in the experimental section comprehensively about the broad scope and use of calcium carbide route. My decision is to reject the present incomplete form of work and resubmit the revised version by including the below concerns.

1. Include the broad scope and significance of calcium carbide route such as nonexplosive, inexpensive and renewable source of acetylene gas etc. in the introduction part by citing the following references (Zhang, *Green Chem.*, 2013, 15, 2718–2721; Ferdi Schuith, *Chem. Rev.* 2014, 114, 1761–1782; Zheng Li, *Eur. J. Org. Chem.*, 2017: 6648-6651; Matsubara, *Org. Lett.* 2015, 17, 2354–2357).
2. The experimental section is not clear about how the authors have generated acetylene gas from calcium carbide. In Page 7, line 38 “The feed gas was only acetylene (99.99%)..” seems like they have used pure acetylene gas directly. In case, they have utilized calcium carbide as a source of acetylene gas via hydrolysis approach using stoichiometric amount of water. They should explain clearly in the experimental section about their experimental set-up and how water or moisture on the catalyst surface influence the outcome results of hydrogenation reaction.
3. Provide a plausible mechanism with including individual role of Pd sites and MCM (SiO₂, Al₂O₃) support for activation of acetylene and hydrogen and selectivity or affinity to adsorb intermediates to afford ethylene selectively. Explain reason for carbon deposition (polymerization side-product) path in the present catalyst.
4. Include XPS analysis for characterization of Pd oxidation state before and after hydrogenation catalysis.
5. Include leaching test experiment to demonstrate the heterogeneity of present catalytic system.
6. The present catalyst has drawback of recyclability due to catalyst poisoning triggered by acetylene polymerization side product.
7. Several grammatical errors persist in the manuscript.

Review form: Reviewer 2

Is the manuscript scientifically sound in its present form?

Yes

Are the interpretations and conclusions justified by the results?

No

Is the language acceptable?

Yes

Do you have any ethical concerns with this paper?

No

Have you any concerns about statistical analyses in this paper?

No

Recommendation?

Accept with minor revision (please list in comments)

Comments to the Author(s)

This manuscript mainly reported the use of ethylene glycol reduction method for the preparation of palladium-based catalysts with different supports for the hydrogenation of acetylene into ethylene. The obtained yield of ethylene could reach 82.87% when using MCM-41 as support, and all the catalysts were well characterized by physical and chemical means. Overall, I would suggest possible publication of the manuscript after some minor changes.

1. In the part of the introduction, it would be better to add explanation for the advantages of calcium carbide in the preparation of acetylene.
2. How is the reusability of the Pd/MCM-41 catalyst? The structure of the used catalyst is also suggested to check and compare with the fresh one.
3. Why Pd/MCM-41 has better catalytic activity than the other two catalysts? The catalyst structure-activity relationship should be discussed properly.
4. How does Pd/MCM-41 interact with acetylene? More explanations are suggested.

Decision letter (RSOS-191155.R0)

12-Aug-2019

Dear Dr Zhu:

Title: Pd/MCM-41 catalyst for acetylene hydrogenation to ethylene via calcium carbide route
Manuscript ID: RSOS-191155

The editor assigned to your manuscript has now received comments from reviewers. We would like you to revise your paper in accordance with the referee and Subject Editor suggestions which can be found below (not including confidential reports to the Editor). Please note this decision does not guarantee eventual acceptance.

Please submit your revised paper before 04-Sep-2019. Please note that the revision deadline will expire at 00.00am on this date. If we do not hear from you within this time then it will be assumed that the paper has been withdrawn. In exceptional circumstances, extensions may be possible if agreed with the Editorial Office in advance. We do not allow multiple rounds of revision so we urge you to make every effort to fully address all of the comments at this stage. If deemed necessary by the Editors, your manuscript will be sent back to one or more of the original reviewers for assessment. If the original reviewers are not available we may invite new reviewers.

To revise your manuscript, log into <http://mc.manuscriptcentral.com/rsos> and enter your Author Centre, where you will find your manuscript title listed under "Manuscripts with Decisions." Under "Actions," click on "Create a Revision." Your manuscript number has been

appended to denote a revision. Revise your manuscript and upload a new version through your Author Centre.

Please also include the following statements alongside the other end statements. As we cannot publish your manuscript without these end statements included, if you feel that a given heading is not relevant to your paper, please nevertheless include the heading and explicitly state that it is not relevant to your work.

- Funding statement

Please include a funding section after your main text which lists the source of funding for each author.

RSC Associate Editor:
Comments to the Author:
(There are no comments.)

RSC Subject Editor:
Comments to the Author:
(There are no comments.)

Reviewers' Comments to Author:
Reviewer: 1

Comments to the Author(s)
Zhu et al. reported Pd/MCM-41 catalyst for acetylene hydrogenation to ethylene via calcium carbide route. However, the hydrogenation of acetylene gas to ethylene using various metal

oxide supported Pd satellite sites are well documented in the literature. The merit of present work is the utilization of in-situ generated acetylene feedstock from an inexpensive and renewable calcium carbide. Although, authors have neither discussed the context in the introduction nor in the experimental section comprehensively about the broad scope and use of calcium carbide route. My decision is to reject the present incomplete form of work and resubmit the revised version by including the below concerns.

1. Include the broad scope and significance of calcium carbide route such as nonexplosive, inexpensive and renewable source of acetylene gas etc. in the introduction part by citing the following references (Zhang, *Green Chem.*, 2013, 15, 2718–2721; Ferdi Schuith, *Chem. Rev.* 2014, 114, 1761–1782; Zheng Li, *Eur. J. Org. Chem.*, 2017: 6648–6651; Matsubara, *Org. Lett.* 2015, 17, 2354–2357).
2. The experimental section is not clear about how the authors have generated acetylene gas from calcium carbide. In Page 7, line 38 “The feed gas was only acetylene (99.99%)..” seems like they have used pure acetylene gas directly. In case, they have utilized calcium carbide as a source of acetylene gas via hydrolysis approach using stoichiometric amount of water. They should explain clearly in the experimental section about their experimental set-up and how water or moisture on the catalyst surface influence the outcome results of hydrogenation reaction.
3. Provide a plausible mechanism with including individual role of Pd sites and MCM (SiO₂, Al₂O₃) support for activation of acetylene and hydrogen and selectivity or affinity to adsorb intermediates to afford ethylene selectively. Explain reason for carbon deposition (polymerization side-product) path in the present catalyst.
4. Include XPS analysis for characterization of Pd oxidation state before and after hydrogenation catalysis.
5. Include leaching test experiment to demonstrate the heterogeneity of present catalytic system.
6. The present catalyst has drawback of recyclability due to catalyst poisoning triggered by acetylene polymerization side product.
7. Several grammatical errors persist in the manuscript.

Reviewer: 2

Comments to the Author(s)

This manuscript mainly reported the use of ethylene glycol reduction method for the preparation of palladium-based catalysts with different supports for the hydrogenation of acetylene into ethylene. The obtained yield of ethylene could reach 82.87% when using MCM-41 as support, and all the catalysts were well characterized by physical and chemical means. Overall, I would suggest possible publication of the manuscript after some minor changes.

1. In the part of the introduction, it would be better to add explanation for the advantages of calcium carbide in the preparation of acetylene.
2. How is the reusability of the Pd/MCM-41 catalyst? The structure of the used catalyst is also suggested to check and compare with the fresh one.
3. Why Pd/MCM-41 has better catalytic activity than the other two catalysts? The catalyst structure-activity relationship should be discussed properly.
4. How does Pd/MCM-41 interact with acetylene? More explanations are suggested.

Author's Response to Decision Letter for (RSOS-191155.R0)

See Appendix A.

RSOS-191155.R1 (Revision)

Review form: Reviewer 1 (Ummadisetti Chinnarajesh)

Is the manuscript scientifically sound in its present form?

Yes

Are the interpretations and conclusions justified by the results?

Yes

Is the language acceptable?

No

Do you have any ethical concerns with this paper?

No

Have you any concerns about statistical analyses in this paper?

No

Recommendation?

Accept with minor revision (please list in comments)

Comments to the Author(s)

The revised version of the manuscript has improved, and the most of the author's responses are supported by the experimental evidence. Still there are unresolved issues such as recyclability of catalyst at the present stage of work which needs to be fixed in the future. Overall, I recommend considering the present work to warrant a publication in RSOS Journal after addressing the following concerns.

Minor Comments

1. The source of acetylene is a calcium carbide mineral, so it can be considered as a renewable substrate. However, the present work does not deal with the direct production of acetylene in situ from a calcium carbide source. The title of present manuscript "Pd/MCM-41 catalyst for acetylene hydrogenation to ethylene via calcium carbide route" is misleading the actual meaning. Authors should delete the phrase "via calcium carbide route" from the title.

2. Still there are typo/grammatical errors persist in the manuscript, for example ...In Page 18, Second paragraph, "As Fig. 4 shown, the reactant acetylene and hydrogen adsorbed on the surface of active Pd sites, and hydrogen dissociated into H atom." Here the typo error "intro should be corrected as into"

Another error with the phrase dipolymer should be written as dimer in the sentence "If no H atom was provided, C₂H₃* would reacted with its dipolymer to produce green oil by side reactions (step 5 and step 6)."

Review form: Reviewer 2

Is the manuscript scientifically sound in its present form?

Yes

Are the interpretations and conclusions justified by the results?

Yes

Is the language acceptable?

Yes

Do you have any ethical concerns with this paper?

No

Have you any concerns about statistical analyses in this paper?

No

Recommendation?

Accept as is

Comments to the Author(s)

The revised manuscript has been improved a lot and may be suitable for publication.

Decision letter (RSOS-191155.R1)

07-Oct-2019

Dear Dr Zhu:

Title: Pd/MCM-41 catalyst for acetylene hydrogenation to ethylene via calcium carbide route
Manuscript ID: RSOS-191155.R1

Thank you for submitting the above manuscript to Royal Society Open Science. On behalf of the Editors and the Royal Society of Chemistry, I am pleased to inform you that your manuscript will be accepted for publication in Royal Society Open Science subject to minor revision in accordance with the referee suggestions. Please find the reviewers' comments at the end of this email.

The reviewers and handling editors have recommended publication, but also suggest some minor revisions to your manuscript. Therefore, I invite you to respond to the comments and revise your manuscript.

Because the schedule for publication is very tight, it is a condition of publication that you submit the revised version of your manuscript before 16-Oct-2019. Please note that the revision deadline will expire at 00.00am on this date. If you do not think you will be able to meet this date please let me know immediately.

To revise your manuscript, log into <https://mc.manuscriptcentral.com/rsos> and enter your Author Centre, where you will find your manuscript title listed under "Manuscripts with Decisions". Under "Actions," click on "Create a Revision." You will be unable to make your

revisions on the originally submitted version of the manuscript. Instead, revise your manuscript and upload a new version through your Author Centre.

Best wishes,
Dr Laura Smith
Publishing Editor, Journals

RSC Associate Editor:
Comments to the Author:
(There are no comments.)

RSC Subject Editor:
Comments to the Author:
(There are no comments.)

Reviewer comments to Author:
Reviewer: 2

Comments to the Author(s)
The revised manuscript has been improved a lot and may be suitable for publication.

Reviewer: 1

Comments to the Author(s)
The revised version of the manuscript has improved, and the most of the author's responses are supported by the experimental evidence. Still there are unresolved issues such as recyclability of catalyst at the present stage of work which needs to be fixed in the future. Overall, I recommend considering the present work to warrant a publication in RSOS Journal after addressing the following concerns.

Minor Comments

1. The source of acetylene is a calcium carbide mineral, so it can be considered as a renewable substrate. However, the present work does not deal with the direct production of acetylene in situ from a calcium carbide source. The title of present manuscript "Pd/MCM-41 catalyst for acetylene hydrogenation to ethylene via calcium carbide route" is misleading the actual meaning. Authors should delete the phrase "via calcium carbide route" from the title.

2. Still there are typo/grammatical errors persist in the manuscript, for example ...In Page 18, Second paragraph, "As Fig. 4 shown, the reactant acetylene and hydrogen adsorbed on the surface of active Pd sites, and hydrogen dissociated intro H atom." Here the typo error "intro should be corrected as into"

Another error with the phrase dipolymer should be written as dimer in the sentence "If no H atom was provided, C₂H₃* would reacted with its dipolymer to produce green oil by side reactions (step 5 and step 6)."

Author's Response to Decision Letter for (RSOS-191155.R1)

See Appendix B.

Decision letter (RSOS-191155.R2)

15-Oct-2019

Dear Dr Zhu:

Title: Pd/MCM-41 catalyst for acetylene hydrogenation to ethylene
Manuscript ID: RSOS-191155.R2

It is a pleasure to accept your manuscript in its current form for publication in Royal Society Open Science. The chemistry content of Royal Society Open Science is published in collaboration with the Royal Society of Chemistry.

RSC Associate Editor
Comments to the Author:
(There are no comments.)

Reviewer(s)' Comments to Author:

Appendix A

Dear Editor,

Thank you for considering our manuscript and for the reviewers' comments. According to comments, the changes in manuscript are marked in red.

We hope that our revised manuscript is acceptable for publication in your journal. Our point-by-point responses to the reviewers' comments are listed below.

Yours sincerely,

Ming-yuan Zhu (Corresponding author),

Tel: +86-993-2057277

Fax: +86-993-2057270

Email: zhuminyuan@shzu.edu.cn (Mingyuan Zhu)

Response to the reviewers:

Reviewer: 1

Comments to the Author(s)

Zhu et al. reported Pd/MCM-41 catalyst for acetylene hydrogenation to ethylene via calcium carbide route. However, the hydrogenation of acetylene gas to ethylene using various metal oxide supported Pd satellite sites are well documented in the literature. The merit of present work is the utilization of in-situ generated acetylene feedstock from an inexpensive and renewable calcium carbide. Although, authors have neither discussed the context in the introduction nor in the experimental section comprehensively about the broad scope and use of calcium carbide route. My decision is to reject the present incomplete form of work and resubmit the revised version by including the below concerns.

1. Include the broad scope and significance of calcium carbide route such as nonexplosive, inexpensive and renewable source of acetylene gas etc. in the introduction part by citing the following references (Zhang, Green Chem., 2013, 15, 2718–2721; Ferdi Schuth, Chem. Rev. 2014, 114, 1761-1782; Zheng Li, Eur. J. Org. Chem., 2017: 6648-6651; Matsubara, Org. Lett. 2015, 17, 2354-2357).

Response: Thank you very much for your valuable suggestion. We have added the presentation of the merits of acetylene from calcium carbide in the introduction part,

and these references were also added.

Corresponding part can be found in page 3 in the revised manuscript.

2. The experimental section is not clear about how the authors have generated acetylene gas from calcium carbide. In Page 7, line 38 “The feed gas was only acetylene (99.99%)..” seems like they have used pure acetylene gas directly. In case, they have utilized calcium carbide as a source of acetylene gas via hydrolysis approach using stoichiometric amount of water. They should explain clearly in the experimental section about their experimental set-up and how water or moisture on the catalyst surface influence the outcome results of hydrogenation reaction.

Response: The acetylene gas used in this work was from gas cylinder directly. It is very difficult to produce the acetylene gas under high pressure by an experimental set-up utilizing calcium carbide. Therefore, we only try to imitate the hydrogen process of acetylene from calcium carbide in the present condition of our laboratory.

We have also corrected the experimental section to make the text more clearly. Corresponding part can be found in page 6.

3. Provide a plausible mechanism with including individual role of Pd sites and MCM (SiO_2 , Al_2O_3) support for activation of acetylene and hydrogen and selectivity or affinity to adsorb intermediates to afford ethylene selectively. Explain reason for carbon deposition (polymerization side-product) path in the present catalyst.

Response: Following your suggestion, we described the adsorption of acetylene and hydrogen reactants, and the reaction mechanism to produce ethylene, ethane and green oil in Fig. 4. Corresponding explanation was also added in page 10 of the revised manuscript.

4. Include XPS analysis for characterization of Pd oxidation state before and after hydrogenation catalysis.

Response: Following your suggestion, we characterize the Pd oxidation state in the fresh and spent catalyst by XPS analysis.

Corresponding part can be found in Fig. 8 and page 16 in the revised manuscript.

5. Include leaching test experiment to demonstrate the heterogeneity of present catalytic system.

Response: Following your suggestion, we detect the leaching of the Pd by comparing the Pd loading in the fresh and spent catalyst using ICP-AES. The actual loading of Pd was 0.096% and 0.094% in the fresh and spent Pd/MCM-41 catalyst, respectively. No obvious Pd leaching was found during the hydrogen process.

Corresponding part can be found in page 16 in the revised manuscript.

6. The present catalyst has drawback of recyclability due to catalyst poisoning triggered by acetylene polymerization side product.

Response: We agree with your viewpoint that this present catalyst has the drawback of recyclability due to the producing of green oil.

The novelty of this work is that Pd catalyst can be used for pure acetylene hydrogenation and that green oil formation is the main reason for low ethylene selectivity and catalyst deactivation. More stable Pd catalyst may be designed and synthesized to meet the criteria of pure acetylene hydrogenation in near future, based on the deactivation mechanism of Pd catalyst.

7. Several grammatical errors persist in the manuscript.

Response: Thanks for your suggestion, we have checked our manuscript very carefully again and correct several grammatical errors.

Reviewer: 2

This manuscript mainly reported the use of ethylene glycol reduction method for the preparation of palladium-based catalysts with different supports for the hydrogenation of acetylene into ethylene. The obtained yield of ethylene could reach 82.87% when using MCM-41 as support, and all the catalysts were well characterized by physical and chemical means. Overall, I would suggest possible publication of the manuscript after some minor changes.

1. In the part of the introduction, it would be better to add explanation for the advantages of calcium carbide in the preparation of acetylene.

Response: Thank you very much for your valuable suggestion. We have added the advantages of acetylene from calcium carbide in the introduction part, and corresponding part can be found in page 3 in the revised manuscript.

2. How is the reusability of the Pd/MCM-41 catalyst? The structure of the used catalyst is also suggested to check and compare with the fresh one.

Response: Following your suggestion, we compared the Pd loading and oxidized status of Pd in the fresh and spent Pd/MCM-41 catalyst. The results were added to page 16 in the revised manuscript.

No Pd metal leaching occurred and only part of oxidized Pd(II) were reduced to metallic Pd (0). So, the main reason for the deactivation of the Pd/MCM-41 catalyst was coke due to the producing of green oil by side reaction. The catalyst can be reuse after removing the coke on the surface of the catalyst, by treating it in H₂ atmosphere in 500°C for 4 h.

3. Why Pd/MCM-41 has better catalytic activity than the other two catalysts? The catalyst structure-activity relationship should be discussed properly.

Response: As Table 1 shown, MCM-41 has higher surface area than that of other two catalysts. Higher surface area of the support will cause Pd dispersed more uniformly in MCM-41, and the results were also confirmed by the TEM results.

Therefore, better uniform dispersion of Pd on the surface of MCM-41 may be one of the reasons for its better C₂H₄ selectivity.

Corresponding part can be found in page10 of the manuscript.

4. How does Pd/MCM-41 interact with acetylene? More explanations are suggested.

Response: Following your suggestion, we described the adsorption of acetylene and hydrogen reactants, and the reaction mechanism to produce ethylene, ethane and green oil in Fig. 4. Corresponding explanation was also added in page 10 of the revised manuscript.

Appendix B

Response to the referees

The revised version of the manuscript has improved, and the most of the author's responses are supported by the experimental evidence. Still there are unresolved issues such as recyclability of catalyst at the present stage of work which needs to be fixed in the future. Overall, I recommend considering the present work to warrant a publication in RSOS Journal after addressing the following concerns.

Minor Comments

Q1. The source of acetylene is a calcium carbide mineral, so it can be considered as a renewable substrate. However, the present work does not deal with the direct production of acetylene in situ from a calcium carbide source. The title of present manuscript "Pd/MCM-41 catalyst for acetylene hydrogenation to ethylene via calcium carbide route" is misleading the actual meaning. Authors should delete the phrase "via calcium carbide route" from the title.

Response: The title of the manuscript has been corrected.

Q2. Still there are typo/grammatical errors persist in the manuscript, for example ...In Page 18, Second paragraph, "As Fig. 4 shown, the reactant acetylene and hydrogen adsorbed on the surface of active Pd sites, and hydrogen dissociated into H atom." Here the typo error "intro should be corrected as into"

Another error with the phrase dipolymer should be written as dimer in the sentence "If no H atom was provided, $C_2H_3^*$ would react with its dipolymer to produce green oil by side reactions (step 5 and step 6)."

Response: Thanks for your carefully review. These errors have been corrected.